# Clinical Outcome of Primary Total Hip Arthroplasty in Patients with Morbid Obesity—Retrospective and Prospective Follow-Up Studies

**DOI:** 10.3390/medicina57111247

**Published:** 2021-11-15

**Authors:** Andrey Gritsyuk, Alexey Lychagin, Liu Yi, Nahum Rosenberg

**Affiliations:** Department of Traumatology, Orthopedics and Disaster Surgery, I.M. Sechenov First Moscow State Medical University, 119992 Moscow, Russia; dr.lychagin@mail.ru (A.L.); wdtcliuyi@gmail.com (L.Y.); nahumrosenberg@hotmail.com (N.R.)

**Keywords:** hip arthroplasty, obesity, surgical complications, hip arthritis, functional outcome

## Abstract

*Background and objective:* There is a general clinical concern on the negative impact of obesity on surgical complications and functional outcomes. We hypothesized that the patients with morbid obesity are exceptionally prone to a significantly increased risk for surgical and short-term complications after primary total hip arthroplasty (THA). We aimed to identify the range of Body Mass Index (BMI) values of patients with a significant risk for lower functional improvement after THA. *Materials and methods:* In Stage 1 of the study, we conducted a retrospective comparative analysis of the rate of complications and functional outcomes in patients treated by primary THA, with normal weight (BMI 19–25, *N* = 1205) vs. Class 1 (BMI 26–34, *N* = 450), Class 2 (BMI 35–39, *N* = 183), and Class 3 (BMI ≥ 40, *N* = 47) obese patients. After the statistical similarity rates of complications and 6- and 12-month functional outcomes (by Harris Hip and SF-36 scores) were revealed in Class 1 patients and patients with normal BMI, we conducted the Stage 2 prospective study, by the same comparison protocol, on the cohorts of Class 2 (*N* = 29) and Class 3 (*N* = 16) patients compared to the Class 1 patients (*N* = 37) as controls. *Results:* Stage 1: There was no difference in surgical complications and function on 6- and 12-month postoperative follow-up (physical and mental) between Class 1 and patients with normal BMI (*p* > 0.05). Surgical complications were significantly higher in Class 2 (*p* < 0.05) and Class 3 (*p* < 0.001) patients. Functional activity on the 12-month follow-up increased significantly in all study groups, but in the Class 3 patients, the functional parameters were significantly lower (0.001). The mental health status on the follow-up was similar in all study groups. Stage 2 study revealed similar to the retrospective study comparison of parameters, except for the significantly lower mental health scores in Class 2 and Class 3 patients (*p* < 0.05) and functional scores in Class 3 patients (*p* < 0.05). *Conclusion:* Although the functional ability increased in all patients, it was significantly lower in Class 3 patients (with morbid obesity). Therefore, the patients with Class 1 and Class 2 obesity should be conceptionally distinguished from Class 3 patients in the decision-making process for a primary THA because of the less favorable functional and mental health improvement in those with morbid obesity (Class 3).

## 1. Introduction

Primary total hip arthroplasty (THA) in patients with morbid obesity is a technically demanding surgical procedure, with substantial risks for early and late postoperative complications [1,2].

There is published evidence that following THA in obese patients, there is an increased risk for postoperative complications [3], but this observation is not substantially supported [4]. Additionally, postoperative rehabilitation might be less successful after primary THA in obese patients due to the inherent difficulty in ambulation.

Studies on long-term prosthesis survivorship and the risk for revision surgery after THA in obese patients have shown that over five years postoperatively, obese patients required revision surgery about 60% more often than patients with an normal weight [5].

Naturally, the uncertainty about the significance of obesity as a risk factor for THA failure should be resolved because this factor can affect the preoperative decision-making process when a bodyweight factor potentially contradicts the clinical necessity for THA. The recommendation has to be based on reliable clinical data, especially in light of the foreseen improvement in postoperative ambulation that might partly resolve the overweight in a sedentary person due to a painful hip.

According to these considerations and our previous clinical observations, we hypothesized that only patients with morbid obesity are expected to suffer from considerable peri- and postoperative complications after primary THA. Therefore, the purpose of this study was to identify the impact of morbid obesity on THA outcome by detecting the range of BMI values of patients with a risk for lower functional improvement after THA.

## 2. Materials and Methods

Two experienced senior orthopedic surgeons did all the surgeries. To narrow the margins of the significant effect of obesity on the surgical outcome, we subdivided the study group patients according to World Health Organization classification of obesity severity, i.e., according to the BMI values: Class 1—30.00–34.99 kg/m^2^; Class 2—35.00–39.99 kg/m^2^; Class 3 (morbid obesity) ≥ 40 kg/m^2^ [6].

We executed this study in two stages. Initially (Stage 1), we aimed to consolidate our clinical impression on the effect of obesity on THA outcome by reviewing the retrospective data on the already treated patients. Then (Stage 2), we have investigated the same parameters by a prospective controlled study. The study was approved by the Institutional Ethical Committee (#129, October 2015), all the patients signed the informed consent form. 

All the evaluated patients were surgically treated for hip joint osteoarthritis (grades 3 and 4 on I. Kellgren and I. Lawrence′s scale [7]), characterized by a pain syndrome of over three points on the visual analog scale (VAS) [8]. All the patients were fit for surgery under general and/or regional anesthesia. In all the patients, we used a porous titanium alloy cup and a titanium alloy stem covered with hydroxyapatite, with metal–polyethylene friction pair (Zimmer^®^, Warsaw, IN, USA, or DePuy^®^, Paramount Drive, MA, USA). These prostheses have similar designs for cementless implantation. The prostheses implantations were performed via the standard anterolateral surgical approach to the hip joint. The perioperative management of patients, surgical technique and type implant, and postoperative and rehabilitation protocols were similar. 

The lengths of the surgeries, perioperative blood loss (expressed by the hemoglobin levels and rate of blood transfusions), early postoperative wound infections (superficial and deep), periprosthetic fractures and joint dislocations, neural surgical damage, and early aseptic loosening [9] of the implanted prostheses were recorded.

The functional activity in all patients was recorded preoperatively and at 6- and 12-month follow-ups by the Harris Hip Scale (HHS) [10]. The hip joint function and the patients′ quality of life were recorded by the SF—36 score [11], preoperatively, and at 12-month follow—up. The HHS is designed to evaluate pain, deformity, and function (including a range of motion) of the hip joint concerning hip surgery, with a 0–100 points scale reflecting an increase in hip joint function. The SF—36 is 36 points questionnaire, with a 0–100 points scale reflecting an increased physical and mental viability.

### 2.1. Stage 1 (Retrospective Study)

#### The Study Groups

In the study period (2010–2015), 2750 patients were treated by the primary THA. In the five years of the postoperative period, 31 patients died (for the unrelate to this surgery reasons), and additional 834 patients were unavailable for the follow-up. Therefore, in the retrospective analysis, we studied the records of 1885 consecutive patients (69% of the initial study group) after primary THA (operated between 2010–2015). The study group included 1205 patients with average body mass index (BMI) and 680 patients with various degrees of obesity (Table 1). These patients were available for a follow—up clinical revaluation at 6 and 12 months postoperatively. The age and gender distributions were similar among the study groups (Table 1). 

### 2.2. Stage 2 (Prospective Study)

#### The Study Groups

Eighty-two consecutive obese (BMI above 30 kg/m^2^) patients (age range 40–85 years, 28 men and 54 women, Table 2) treated by primary THA in the years 2016–2017 were evaluated prospectively by the same criteria and the protocol that we implemented in Stage 1 of this study. 

Seven patients with higher than 30 kg/m^2^ BMI but younger than 40 years or older than 85 were excluded from the study, aiming to avoid additional unrelated functional age-related factors that might affect the postsurgical outcome. All the patients in this study group were available for the follow-up evaluation.

Accordingly, the study group consisted of 16, 29, and 37 patients of Classes 1, 2, and 3 of obesity. The distribution of ages was similar in all study groups; the gender distribution was similar in Classes 2 and 3. There were no male patients in the Class I group (Table 2). 

### 2.3. Statistics 

The evaluation of the functional outcome of THA with a statistical power level of 80% (with an α level of 0.05) requires at least 65 patients [12,13]. The present report meets these statistical power requirements for meaningful outcome interpretation.

The results are presented as average values with indication of standard error of mean (SEM).

The Chi-square test and Fisher′s exact test (when over 25% of cells had less than five cases) were used in comparing categorical data. Independent *t*-tests compared the normally distributed continuous variables for unpaired variables, a paired *t*-test for paired (matched) variables, and a one-way ANOVA for more than two variables. We compared nonparametric data using the Mann–Whitney test. For all statistical tests, we set the level of statistical significance at *p* < 0.05.

We used the SPSS Statistics 22.0 statistical software (SPSS Inc., Chicago, IL, USA). 

## 3. Results

### 3.1. Stage 1 (Retrospective Study)

The surgery time was similar for patients with Class 1 obesity and patients with a normal BMI (66.4 ± 0.63 SEM min vs. 64.4 ± 0.4 SEM min, *p* = 0.68). Surgery time in patients with Class 2 and 3 obesity was significantly longer in comparison to the patients with normal BMI (76.9 ± 1.18 SEM min,19.4% longer, *p* = 0.04, 87.4 ± 2.41 SEM min 26.3% longer, *p* = 0.002, respectively vs. 64.4 ± 0.4 SEM min).

There was no difference in the preoperative blood hemoglobin levels (13.5–13.8 g/dL range, *p* = 0.558). Twenty-four hours postoperatively, we found a significantly higher drop in hemoglobin content in Class 2 and Class 3 patients in comparison to the patients with normal BMI (mean 3.2 g/dL and 4.6 g/dL respectively vs. 1.6 g/dL, *p* = 0.036), without significant difference among Class 1 patients in comparison to the non-obese patients. Similarly, the rate of the requirement for allogenic erythrocyte transfusions was significantly higher in Class 2, and Class 3 patients in contrast to patients with normal BMI (4.6% and 18% of patients respectively vs. 3.9% of patients, *p* = 0.048), and there was the same blood transfusion rate in Class 1 and non-obese patients.

Preoperatively there was no significant difference in HHS scores among all the groups of patients (range of mean scores 44.2–48.0, *p* > 0.05, Figure 1). The assessment of HHS scores at the follow-up showed that in all study groups, the function gradually and significantly improved in the affected extremity over a year after surgery (*p* < 0.01 in all groups, Figure 1). The functional scores dynamics profile was significantly higher in patients with a normal BMI than in patients with morbid obesity (Class 3) (mean score 88.9 ± 0.2 SEM vs. 84.8 ± 0.7 SEM, *p* = 0.0028). There was no significant difference in scores′ profiles of Class 1 and 2 patients compared to the patients with normal BMI (*p* > 0.05, Figure 1).

In all the examined patients, there was a significant increase in physical functioning and mental health scores according to the SF—36 scale (in the range of 40–50% in physical functioning, *p* < 0.001; Figure 2 and in the range of 70–80% in the scores of the mental health component *p* < 0.0001). The normal BMI patients′ group exhibited significantly better dynamics of improvement in the physical constituent of health a year after surgery in comparison to Class 3 patients (mean 28.7 scores ± 0.1 SEM vs. 24.1 ± 0.3 SEM, *p* < 0.05, Figure 2). There was no significant difference in the SF-36 functional scores among Class 1 and 2, and patients with normal BMI. (*p* > 0.05, Figure 2).

There was no significant difference in the mental health component of the SF-36 scale in all study groups (*p* > 0.05, Figure 3).

The total postsurgical complications rate was low in patients with normal BMI (0.15%) and Class 1 obesity (0.25%), with considerably higher and rising rates from the Class 2 up to the Class 3 patients (1.3%, 2.4%, and 4.1% respectively). The rates of postsurgical infections (deep and superficial), prosthesis dislocations, thromboembolic and neurological complications were 2–3 times higher in patients with Class 2 and Class 3 obesity in comparison to patients with Class 1 obesity and normal BMI (*p* < 0.05, Table 3). Early aseptic loosening and prosthesis component wear were not observed at the 12-month follow-up in Class 1 and with normal BMI patients. Among Class 2 or Class 3 patients, these complications occurred at 1.7%, 0.6%, and 0.2%, 0.4% respectively. Similarly, periprosthetic fractures occurred only in Class 2 and 3 patients at 0.15% and 0.3%, respectively (Table 3). Neural damage did not occur in patients with normal BMI but was observed in one patient (0.05%) in Class 1 and Class 2 groups and three patients in the Class 3 group (0.15%).

After reviewing the results of the Stage 1 study, it has become clear that there is no significant difference in the investigated clinical parameters between patients with a normal BMI and patients with Class I obesity in the perioperative period and at the 12-month follow-up; therefore, in the Stage 2 study, we related to the Class 1 patients as a reference group for the clinical comparison to the Class 2 and Class 3 patients.

### 3.2. Stage 2 (Prospective Study)

The length of the surgery in the Class 3 patients was significantly longer (in 26%) than in patients of the Class 1 group (*p* = 0.0032) but was not different in comparison to patients of the Class 2 group (*p* = 0.06). 

There was no difference in the preoperative blood hemoglobin levels in all patients (12.2–14.6 g/dL range, *p* = 0.558). Twenty-four hours postoperatively, we found a significantly higher drop in hemoglobin content in Class 3 and Class 2 patients than the Class 1 patients (mean 4.8 g/dL and 3.2 g/dL respectively vs. 1.6 g/dL, *p* = 0.036)). There was no significant difference in the required rate for allogeneic erythrocyte transfusions in Class 1, Class 2, and Class 3 patients (18%, 20%, and 24% of patients respectively, *p* = 0.442). 

Preoperatively there was no significant difference in HHS scores among all the groups of patients (range of mean scores 44.2–49.5, *p* > 0.05, Figure 4). The assessment of HHS scores at the follow-up showed that, in all study groups, the function gradually and significantly improved in the affected extremity over a year after surgery (*p* < 0.01 in all groups, Figure 4). The outcome analysis using the HHS showed that the Class 3 patients had significantly lower scores than Class 1 and 2 patients at 12-month follow-up (mean 78.1 vs. 86.2 and 84.8 respectively, *p* < 0.01) and at the shorter six-month follow—up (60.2 vs. 64.8 and 68.3 respectively, *p* < 0.05, Figure 4)). There was no difference in scores among Class 1 and Class 2 patients at the follow-up (*p* > 0.05).

Preoperatively in all the examined patients, there was no significant difference in physical functioning and mental health scores according to the SF—36 scale (range of mean values of 11.8–13.4 and 6.8–8.1 respectively, *p* > 0.05, Figure 5). In all the examined patients, there was a significant increase in physical functioning and mental health scores according to the SF—36 scale (in the range of 51–52% in physical functioning, *p* < 0.001; and in the range of 70–78% in the scores of the mental health component *p* < 0.0001; Figure 6). The Class 1 patient group exhibited statistically significantly better result dynamics in the physical constituent of health after a year following the surgery than the Class 3 patients (mean 27.3 score ± 0.3 SEM vs. 24.5 ± 0.5 SEM, *p* < 0.05, Figure 6). There was no significant difference in the SF-36 functional scores among Class 1 and Class 2 patients (*p* > 0.05, Figure 6). There was no significant difference in the mental health component of the SF-36 scale in Class 2 and 3 patients (*p* > 0.05), but these scores were significantly lower in comparison to Class 1 (*p* < 0.05, Figure 6).

The total postsurgical complications rate was significantly lower in patients with Class 1 obesity (1.2%) compared to considerably higher and rising rates from the Class 2 to the Class 3 patients (4.8% and 17.1% respectively, *p* = 0.001, Table 4). The rate of superficial postsurgical infections was significantly higher in patients with Class 3 obesity than the Class 1 patients. These rates raised also in Class 2 patients but especially in Class 3 patients (2.4% and 3.6% respectively, *p* < 0.05, Table 4). The prosthesis dislocations, late aseptic loosening, and periprosthetic fractures occurred only in the Class 3 group (2.4%, 2.4%, and 3.6%, respectively). Neural damage did not appear in Class 1 patients but in one patient (1.2%%) in each Class 2 and Class 3 group.

## 4. Discussion

This study evolved from the initial clinical impression that only extremely obese patients are expected to significantly increase the risk for surgical and short-term complications after primary THA. Cleary this impression is of high importance since there is great difficulty in reducing the body weight in obese patients with symptomatic osteoarthritis because the hip pain impairs their ability to ambulate. Additionally, preventing surgical treatment in all obese patients, because of the expected increased risk for complications, retains high personal cost in terms of quality of life. Therefore, we tried to narrow the obesity parameters (BMI) range in relation to the highest probability of increased surgical and postsurgical complications. 

Recent large scale systematic review [14] intended to clarify this subject and showed, according to the studies on 66,238 THA surgeries from 16 reports, that the rate of complications in the combined group of Class 2 and 3 patients is increased only "slightly" in comparison to patients with a normal BMI. Furthermore, there is an additional retrospective study on a large cohort of patients (1565 patients) after primary THA, when Class 2 and 3 patients were considered as one group and were compared to the patients with normal BMI, that showed an increased rate of peri-surgical complications (longer operation time, higher blood loss and longer hospitalization) [3]. However, in another report on 83146 patients, when the Class 2 and Class 3 patients were evaluated separately, only Class 3 patients were prone to increase perioperative and postoperative complications, mainly due to wound infections [15]. This large-scale published data indicates the necessity to separate the Class 2 and Class 3 patients in predicting the effect of BMI on the rate of surgical complications and short-term functional outcomes following primary THA.

The results of the present study substantiate this important but previously unresolved concern. In the Stage I retrospective large-scale study, we clearly show that the Class 1 patients are identical in the prediction of perioperative and short-term complications with patients with normal BMI (Table 5). According to this study, Class 2 patients had an increased risk for perioperative complications (blood loss, wound infection, periprosthetic fractures, and prosthetic dislocations, with longer surgery time) but were not different in functional scores (HSS and SF-36) at the 12-month follow-up. However, the functional scores in these patients were less favorable at the six-month follow-up. From this data, we could deduce that Class 2 patients reach the same functional ability as patients with normal BMI, but with a lower slope of the intermediate recovery rate and increased risk for perioperative complications. Naturally, this observation can be explained by more complicated tissue handling in more obese patients during surgery.

As hypothesized, the Class 3 morbidly obese patients have a significantly higher rate of perioperative complications and decreased functional abilities (HSS and functional component of the SF-36 score). However, since these patients improved in their function to some extent following the surgery and their mental health scoring ( by the SF-36 mental component) reflected similar to the not obese (with normal BMI) patients improvement at 12-month follow-up (Table 5).

Thus, we clearly showed the difference in perioperative complications rate and short term postoperative function ability between Class 2 and Class 3 patients that indicates the necessity for the different preoperative decision-making processes, with a clear indication that mildly obese Class 1 patients (below BMI of 35) should be treated as patients with normal BMI.

The Stage 2 study was designed to consolidate these results by evaluating the observed perioperative and short-time follow-up difference in Class 2 and Class 3 patients′ surgical outcomes by implementing similar to the Stage 1 study protocol, prospectively.

The main finding of this study, according to the prospective study results that substantially support the indication from the retrospective study, is a significant difference in the rate of surgical complications and functional ability, at the 12-month follow-up, with more favorable outcomes in Class 2 patients vs. Class 3 patients. Interestingly, as opposed to the Stage 1 study, the mental health scores were similarly reduced in both groups, and there was no increase in blood transfusion rates in both groups. However, in both, there was significantly higher blood loss. The later facts might be attributed to the recent more conservative approach for blood transfusion indications (in comparison to previous ones that were implemented in the retrospective evaluation) and higher expectations for functional improvement in the later period of the prospective study.

Therefore, because there is an apparent difference in complications and outcomes between Class 2 and Class 3 patients and the preoperative weight reduction in obese patients is a highly complicated process, the necessity of preoperative attempts for weight loss should be directed primarily to the Class 3 patients with morbid obesity. The Class 2 patients, although they should be encouraged to lose weight, should not be rejected for the primary THA only based on the obese weight, because the significantly improved ambulation ability that is expected after the surgery might facilitate the weight loss and subsequentially reduce the risk of prosthetic wear due to high body weight. Unfortunately, this argument is less implacable to patients with morbid obesity (Class 3); therefore, these two groups of obese patients should be distinguished at the preoperative decision making. 

The main limitation of this retrospective (Stage 1) study is the 31% rate of patients available for the follow-up, but this relative disadvantage is “compensated” by the results of the prospective (Stage 2) study that shows substantially similar results. Additional important limitation is the inherent bias of the scoring recordings at the follow up.

## 5. Conclusion

From these studies, substantial conclusions can be raised. Patients with a BMI up to 35 should be considered to have the lowest rate of perioperative complications and are expected to have favorable short-term functional and mental outcomes. Patients with BMI above 35 have an increased risk for perioperative complications, but this risk is much higher and distinguishable in Class 3 morbid patients (BMI above 40). Although an increase was observed in all patients, the functional outcome improvement after THA is significantly lower in Class 3 patients.

## Figures and Tables

**Figure 1 medicina-57-01247-f001:**
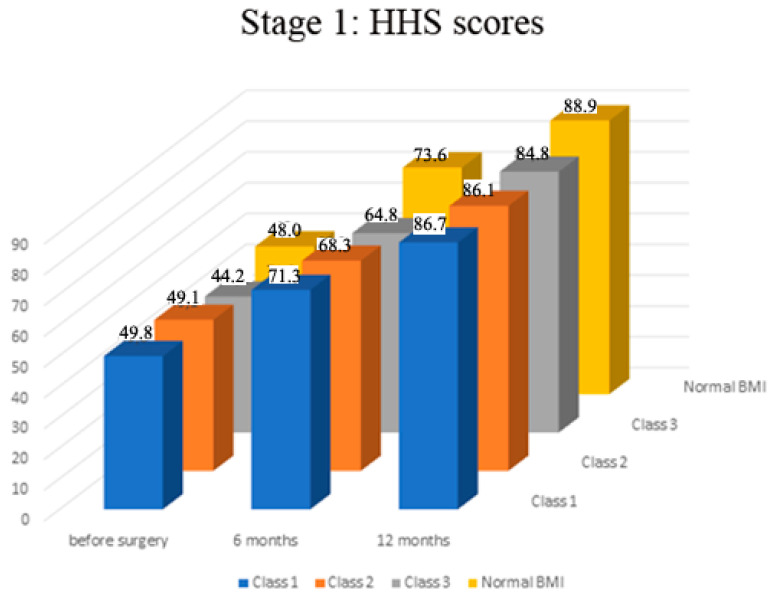
Stage 1—dynamics of functional results by Harris hip score (HHS) before and after surgery.

**Figure 2 medicina-57-01247-f002:**
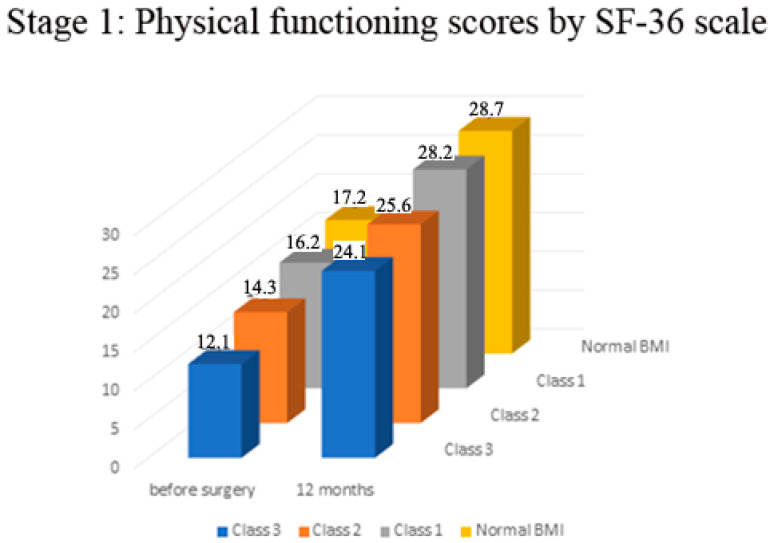
Stage 1—Physical functioning before and 12 months after surgery.

**Figure 3 medicina-57-01247-f003:**
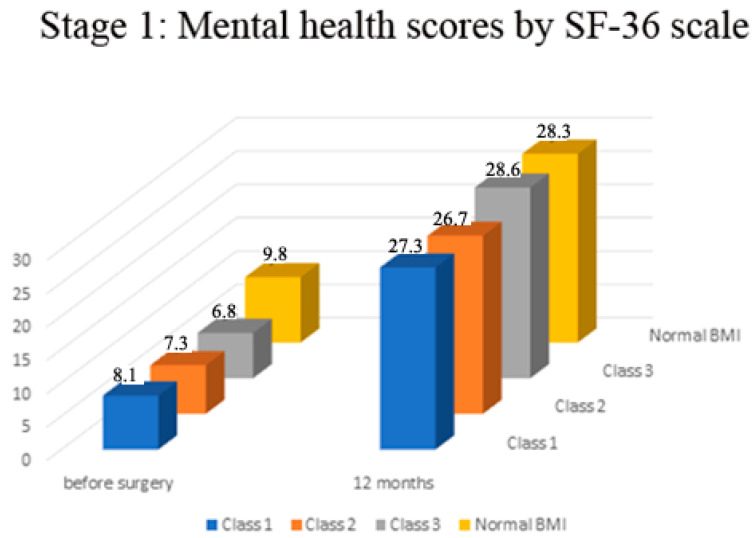
Stage 1—Mental health status before and 12 months after surgery.

**Figure 4 medicina-57-01247-f004:**
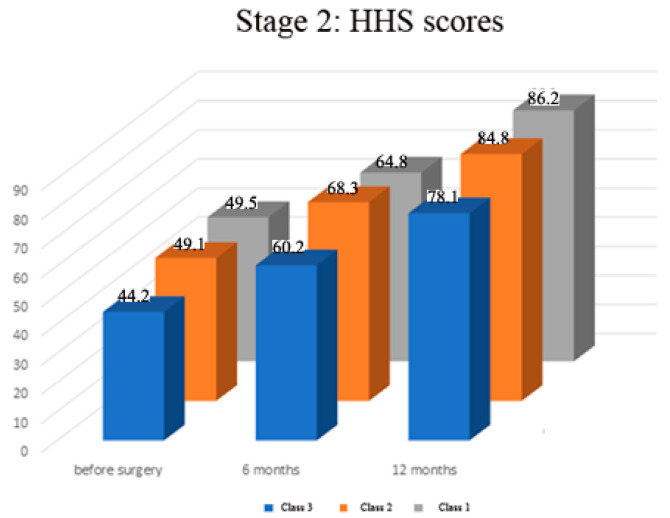
Stage 2—Dynamics of functional results by Harris hip score (HHS) before and after surgery.

**Figure 5 medicina-57-01247-f005:**
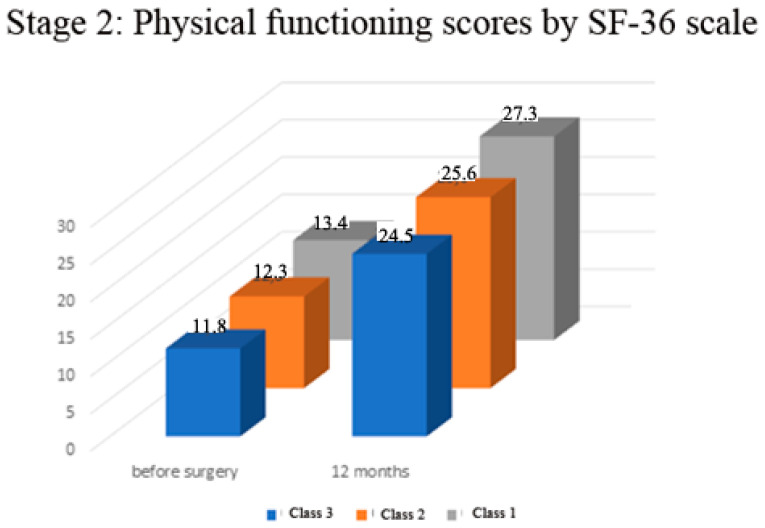
Stage 2—Physical functioning before and 12 months after surgery.

**Figure 6 medicina-57-01247-f006:**
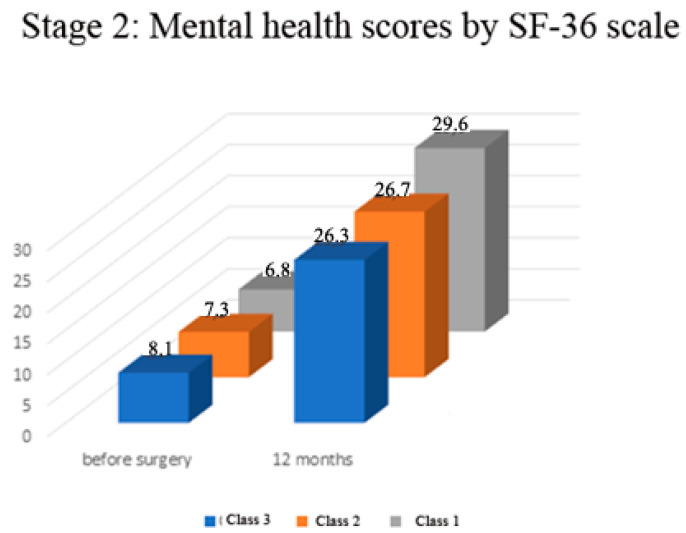
Stage 2—Mental health status before and 12 months after surgery.

**Table 1 medicina-57-01247-t001:** Stage 1 of the study—patient demographics.

Demographics	Groups				*p*-Value
	Normal BMI	Class 1	Class 2	Class 3	
*N* (%)	1205 (63.9%)	450 (23.9%)	183 (9.7%)	47 (2.5%)	0.583
Age * (y)	69.1 ± 2.3	62.3 ± 3.2	61.0 ± 1.9	58.2 ± 1.3	0.927
BMI * (kg/m^2^)	22.6 ± 2.5	32.9 ± 2.9	37.8 ± 2.8	43.9 ± 3.3	0.568
Gender ^#^: Male/ Female	550/655	152/298	84/99	19/28	0.962

y—years, BMI—body mass index; * Analyzed using the one-way ANOVA; ^#^ Analyzed using the Pearson chi-square or the Fisher exact test.

**Table 2 medicina-57-01247-t002:** Stage 2 of the study—patient demographics.

Demographics	Total	Groups			*p*-Value
		Class 1	Class 2	Class 3	
*N* (%)	82 (100%)	16 (19.5%)	29 (35.4%)	37 (45.1%)	0.572
Age * (y)	60.2 ± 2.3	58.2 ± 1.3	61.0 ± 1.9	62.3 ± 3.2	0.981
BMI * (kg/m^2^)	-	32.2 ± 1.1	37.8 ± 1.2	43.9 ± 2.5	0.512
Gender ^#^: Male/ Female	28/54	0/16	13/16	15/22	0.977

y—years, BMI—body mass index; * Analyzed using the one-way ANOVA; ^#^ Analyzed using the Pearson chi-square or the Fisher exact test.

**Table 3 medicina-57-01247-t003:** Stage 1 of the study—complications.

Complications	Normal BMI (*n* = 1205)		Class 1 (*n* = 450)		Class 2 (*n* = 183)		Class 3 (*n* = 47)		Total (*n* = 1885)		*p*-Value
	*n*	%	*n*	%	*n*.	%	*n*	%	*n*	%	
Surface surgical site infection	1	0.05	1	0.05	3	0.15	5	0.25	10	0.5	0.031
Deep surgical site infection	1	0.05	2	0.1	3	0.15	4	0.2	10	0.5	0.043
Periprosthetic fractures	-		-		3	0.15	6	0.3	9	0.47	0.005
Aseptic loosening	-		-		7	1.72	12	0.6	19	1.0	0.046
Prosthesis component wear	-		-		4	0.2	8	0.4	12	0.6	0.326
Thrombophlebitis, thromboembolism of the pulmonary artery	-		1	0.05	-		2	0.1	3	0.2	0.011
Prosthesis dislocations	1	0.05	-		3	0.15	5	0.25	9	0.47	0.016
Neural disorder	-		1	0.05	1	0.05	3	0.15	5	0.5	0.056
Total	3	0.15	5	0.25	24	1.3	45	2.4	77	4.1	0.002

**Table 4 medicina-57-01247-t004:** Stage 2 of the study—complications.

Complications	Class 1 (*n* = 16)		Class 2 (*n* = 29)		Class 3 (*n* = 37)		Total (*n* = 82)		*p*-Value
	abs.	%	abs.	%	abs.	%	abs.	%	
Surface surgical site infection	1	1.2	1	1.2	3	3.6	5	6.0	0.03
Deep surgical site infection	-	-	2	2.4	3	3.6	5	6.0	0.04
Periprosthetic fractures	-	-	-	-	3	3.6	3	3.6	-
Aseptic loosening	-	-	-	-	2	2.4	2	2.4	-
Prosthesis dislocations	-	-	-	-	2	2.4	2	2.4	-
Neural disorder	-	-	1	1.2	1	1.2	2	2.4	0.05
Total	1	1.2	4	4.8	14	17.1	19	23.2	0.001

abs: Absolute.

**Table 5 medicina-57-01247-t005:** Descriptive summary of the comparison of clinical outcome of patients with obesity Class 1, 2, 3 and with normal BMI in the retrospective (Stage 1) study(A) and of patients with obesity Class 2, 3 in comparison to patients with Class 1 obesity in the prospective (Stage 2) study. ↑—significantly increased. ↓—significantly decreased. ↔—no significant difference.

**A (Stage 1)**			
**Clinical Parameters**	**Class 1**	**Class 2**	**Class 3**
Surgery time	↔	↑	↑
Blood loss/transfusion	↔	↑	↑
Complications	↔	↑	↑↑
HHS	↔	↔	↓
SF—36 functional	↔	↔	↓
SF—36 mental	↔	↔	↔
**B (Stage 2)**			
**Clinical Parameters**	**Class 2**		**Class 3**
Surgery time	↔		↑
Blood loss	↑		↑
Blood transfusion	↔		↔
Complications	↑		↑↑
HHS	↔		↓
SF—36 functional	↔		↓
SF—36 mental	↓		↓

## Data Availability

Data is available for the review, kept in the Department of traumatology, orthopedics and disaster surgery, I.M. Sechenov First Moscow State Medical University, Moscow, Russian Federation.

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
