# Peer review of "Clinical Outcome of Primary Total Hip Arthroplasty in Patients with Morbid Obesity—Retrospective and Prospective Follow-Up Studies"

_medicina, 2021, doi:10.3390/medicina57111247_

Round 1
Reviewer 1 Report
Title is clear and analyze type of study
Abstract: Aim is not reported
Methods: hard to understand what you mean for stage 1, stage 2...
Informations about statistical analysis are missing
Results: well reported
Conclusions: coherent with the study
Introduction
Well written. you started with the topic without not necessary paragraphs
Add clear purpose of the study
Methods:
were strobe checklist followed for case series?
did patients signed informed consent for the study?
inclusion and exclusion criteria are not clear
what type or THA was implanted? dual mobility? single mobility? cemented? cementless?
statistics analysis is ok
results
really confusing to analyze first retrospective, than prospective.
results section is really too long
discussion
start with main findings of the study
analyze controversies presented in literature.
report a comparison with present literature
limitations are much more than those reported
conclusions too long. don't split conlcusions in subheadings
Author Response
Abstract: Aim is not reported:
We added aims in the abstract
Methods: hard to understand what you mean for stage 1, stage 2...
Given in the Methods: “We executed this study in two stages. Initially (Stage 1), we aimed to consolidate our clinical impression on the effect of obesity on THA outcome by reviewing the retrospective data on the already treated patients. Then (Stage 2), we have investigated the same parameters by a prospective controlled study.
Informations about statistical analysis are missing”:
Given in the Statistics section: “The evaluation of the functional outcome of THA with a statistical power level of 80% (with an α level of 0.05) requires at least 65 patients [12,13]. The present report meets these statistical power requirements for meaningful outcome interpretation.
The results are presented as average values with indication of standard error of mean (SEM).
The Chi-square test and Fisher's exact test(when over 25% of cells had less than five cases) were used in comparing categorical data. Independent t-tests compared the normally distributed continuous variables for unpaired variables, a paired t-test for paired (matched) variables, and a one-way ANOVA for more than two variables. We compared nonparametric data using the Mann-Whitney test. For all statistical tests, we set the level of statistical significance at p < 0.05.
We used the SPSS Statistics 22.0 statistical software (SPSS Inc., Chicago, Illinois).
Introduction
Add clear purpose of the study:
We added the clarification of the purpose of the study at the end of the Introduction
Methods:
were strobe checklist followed for case series?
Yes and all its sections are reflected in this report: Title, Abstract ,Introduction (Background, Objectives, hypothesis etc.) Methods (Study design settings, patients, variables statistical analysis ),Results (including demographic data, outcome data), Discussion (key results and their interpretation).
did patients signed informed consent for the study?
Yes. This is stated in the second paragraph of the Material and methods section
inclusion and exclusion criteria are not clear
Given in the “Study groups” section. All the patients who were treated by the THA were included. Those who were not available for the follow up were subsequentially excluded in the retrospective study. And all the consecutive patients with no exclusion in the prospective study. This is clearly stated in quantified in the text.
what type or THA was implanted? dual mobility? single mobility? cemented? cementless?
The information is given in the the Material and methods section; “ In all the patients, we used a porous titanium alloy cup and a titanium alloy stem covered with hydroxyapatite, with metal-polyethylene friction pair (Zimmer® or DePuy®). These prostheses have similar designs for cementless implantation.”
results
really confusing to analyze first retrospective, than prospective.
The rationale was to clarify the initial clinical impression in the retrospective study and after it has been substantialized , to give a highly reliable conclusion in the prospective study. This approach is given and described in the methods section: “We executed this study in two stages. Initially (Stage 1), we aimed to consolidate our clinical impression on the effect of obesity on THA outcome by reviewing the retrospective data on the already treated patients. Then (Stage 2), we have investigated the same parameters by a prospective controlled study. “
results section is really too long
In the Results section we provide all the data with the statistical analysis. The data is extensive, according to the study design, and should be fully provided and described. There is no irrelevant parts in the results presentation. To simplify the presentation of a large volume of data we summarized it in the Table 5
discussion
start with main findings of the study
We added the clarification in the 3rd from the last paragraph of the Discussion section
analyze controversies presented in literature.
Given in the 2nd paragraph of the Discussion
report a comparison with present literature
Given in the 2nd paragraph of the Discussion
limitations are much more than those reported
We added the main limitations of the studies at the end of the Discussion.
conclusions too long. don't split conlcusions in subheadings
We removed the subheadings and shortened the text in the Conclusion section
Reviewer 2 Report
This study revealed a poor result in morbid obesity patients with primary THA in one institution. The results were clearly presented. However, several data are necessary; diagnosis of the hip disease, comorbidites before surgery. And also, length of stay, follow-up periods are required. Please spell fully "SEM".
Author Response
diagnosis of the hip disease:
This information is given in the Methods – “treated for hip joint osteoarthritis (grades 3 and 4 on I. Kellgren and I. Lawrence's scale [7]), characterized by a pain syndrome of over 3 points on the visual analog scale (VAS) [8]. “
comorbidites before surgery:
We added that “”All the patients were fit for surgery under general and/or regional anesthesia”
And also, length of stay:
We added the information in the paragraph four in the Methods
follow-up periods are required:
This information is given in the Methods : “six months and 12 months postoperatively”
Please spell fully "SEM".:
We added the full term of standard error of mean (SEM) in the Statistics section
Round 2
Reviewer 1 Report
authors answered in full to all my queries